# Review on Application of Quaternary Ammonium Salts for Gas Hydrate Inhibition

**Haizatul Hafizah Hussain and Hazlina Husin \*** 

Department of Petroleum Engineering, Universiti Teknologi PETRONAS,
Bandar Seri Iskandar 32610, Perak, Malaysia; haizatulhafizah.hus@utp.edu.my
**\*** Correspondence: hazlina.husin@utp.edu.my

**Abstract:** Gas hydrate solids occurrence is considered as one of the serious challenges in flow assurance as it affects the hydrocarbon production significantly, especially in deep water gas fields. The most cost-effective method to inhibit the formation of hydrate in pipelines is by injecting a hydrate inhibitor agent. Continuous studies have led to a comprehensive understanding on the use of low dosage hydrate inhibitors such as ionic liquid and quaternary ammonium salts which are also known as dual function gas hydrate inhibitors. This paper covers the latest types of quaternary ammonium salts (2020–2016) and a summary of findings which are essential for future studies. Reviews on the effects of length of ionic liquids alkyl chain, average suppression temperatures, hydrate dissociation enthalpies, and electrical conductivity to the effectiveness of the quaternary ammonium salts as gas hydrate inhibitors are included.

**Keywords:** gas hydrates; hydrates inhibition; quaternary ammonium salts

## 1. Introduction

In developing deep water gas fields (about 3000 m water depth), the development-gathering mode of "drilling platform-underwater production submarine pipelines" is often implemented [1]. The submarine pipelines are the crucial components that provide transportation means of natural gas to flow from the reservoir wellbore to different types of deep water floating platforms such as large multi-functional semi-submersible platform (Semi-FPS), tension leg platform (TLP) [2], compliant piled tower (CPT) and deep-draft single column platform (Spar). Due to low temperatures, high static water pressure, long tieback distance, and the composition of the gas, gas hydrate solids will possibly occur in riser, subsea trees or subsea pipeline. Gas hydrates are ice-like, crystalline compounds of gas and water [3] that exist at a suitable range of high pressures and low temperatures. It consists of variety of gas molecules such as methane, ethane, propane, isobutene, n-butane, nitrogen, carbon dioxide, hydrogen sulphide, etc. [4]. Gas molecules, known as 'guest' are entrapped in the hydrogen-bonded water molecules, called 'host' [5]. Among the unique features of hydrates are; they are not chemical compounds and are non-stoichiometric crystals [6,7]. Due to no strong chemical bonds exist between the gas and water molecules [8], there is potential of the gas molecules to be released from the water molecules cage [9], which contributes to the flow assurance issues. The three most common gas hydrates structures are structure I (sI), structure II (sII), and structure H (sH), which differs in cage size and physical shape. Typically, structure I consists of small hydrate lattices that can only hold small gas molecules such as $CH_4$. Structure II is more complex, quite larger in size, and able to entrap larger hydrocarbon molecules, whereas structure H is capable to contain much larger molecules such as isopentane [10]. Figure 1 shows the unit cells of sI, sII, and sH structures which are composed of different types of water cages. The conditions of gas hydrate solids occurrence is majorly governed by either its thermodynamic equilibrium or nucleation/growth rate. Chemical

injection by adding the thermodynamic hydrate inhibitors (THIs) like methanol and mono-ethylene glycol (MEG) [5] or low-dosage hydrate inhibitors (LDHIs) such as kinetic hydrate inhibitors (KHIs) and anti-agglomerants (AAs) [11] is a common approach to either shift the equilibrium curve or delay the nucleation rate [12–17].

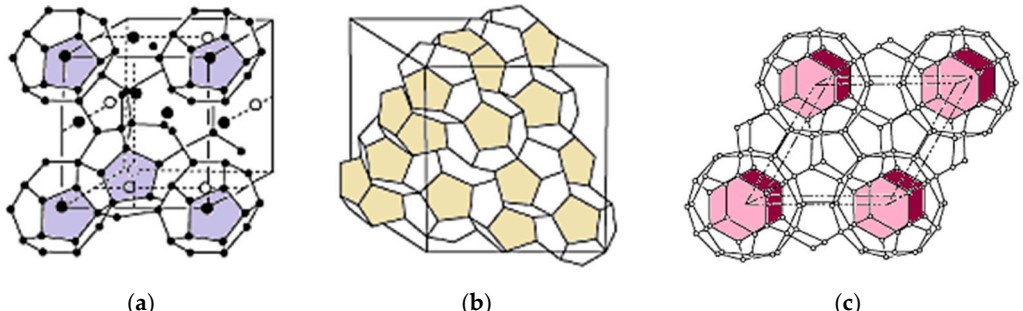

| (**a**) | (**b**) | (**c**) |

**Figure 1.** The three common gas hydrate structures with assembling of unit cells. (**a**) Structure I, (**b**) Structure II and (**c**) Structure sH.

## 2. Flow Assurance Issues Related to Natural Gas

Flow assurance is a term used to evaluate the effects of fluid hydrocarbon solids such as hydrate, wax, and asphaltene; and their potential to disrupt production due to deposition in the pipeline system [18,19]. Deep water environment that exists at low temperatures and high pressures provides the ideal condition for solid deposits such as hydrates to form, with the risk of reducing the hydrocarbon production. Essentially, in petroleum production operations, flow assurance activities ensure operational and economical sustainability of hydrocarbon streams from the reservoir to the surface [20]. Hydrates are often regarded as one of the most serious and challenging problems in flow assurance, since the formation rate of gas hydrates are relatively more rapid than other solid deposits, such as asphaltenes, scale and wax [21]. Hydrates which are formed inside the subsea flowlines affect the production of hydrocarbon significantly, since the formation creates blockage which seriously affect the development and production safety of deep water gas fields [22–25]. Gas-dominated wells are prone to hydrate blockage since the system cool more rapidly compared to the oil-dominated wells, which are typically insulated by design to sustain its high temperature in the flowlines prior to arrival to the surface [26]. Furthermore, when the formed hydrate completely restricts the pipelines, the pressure contained inside the pipelines will increase sharply and eventually causes a serious pipeline safety accident [27,28]. In order to ensure undisturbed flow of hydrocarbon transportation to the surface, up to 8% of the total estimated cost, which is equivalent to more than USD200 million is spent annually on gas hydrate inhibition techniques, as reported by [29].

## 3. Gas Hydrate Inhibition

Gas hydrates can be inhibited by either mechanical methods, i.e., pipe insulation, dehydration, and depressurization, or by chemical methods through injecting special chemicals called as hydrate inhibitors into pipelines [30]. Mechanical methods are considered impractical since they show disadvantages, such as dehydration is impossible between the well and the dehydration units, pipe insulation is too difficult and expensive to be implemented in the deep sea, and depressurization results in reduced transportation capability [31]. Thus, the most practical and economical method is by injection of hydrate inhibitor. In general, there are two major classifications of hydrate inhibitors, which are thermodynamic hydrate inhibitors (THIs) and low dosage hydrate inhibitors (LDHIs).

### 3.1. Thermodynamic Hydrate Inhibitors (THIs) and Low Dosage Hydrate Inhibitors (LDHIs)

Conventional THIs are usually based on anti-freezing solvents like methanol and mono ethylene glycol (MEG) [32], which principally works by shifting hydrate liquid vapor equilibrium (HLVE),

however once the hydrate formation initiates, they facilitated the hydrate nucleation process and demonstrated kinetic promoters characteristics [20,33]. Meanwhile, LDHIs are generally classified into two types, kinetic hydrate inhibitors (KHIs) and anti-agglomerants (AAs). The inhibition mechanism of LDHIs differs from THIs. The inhibition by KHIs is achieved by delaying the nucleation process of the hydrate formation [34], whereas AAs allow hydrate formation however avoid agglomeration. Typically, KHIs are water-soluble polymers like polyvinyl pyrrolidone (PVP) and polyvinyl caprolactam (PVCap) [35]. Figure 2 shows a schematic diagram of kinetic hydrate inhibition mechanisms via adsorption and perturbation [36]. Adsorption inhibition shows that inhibitor molecules are adsorbed on the hydrate crystals, whereas perturbation inhibition works by disturbing the water molecules' structure. In the past decade, ionic liquids (ILs), amino acids (AACs) and quaternary ammonium salts (QAS) [37] have been introduced as green hydrate inhibitors, as well as dual function inhibitors, because they are environmentally friendly and have shown good inhibitory performance as both THIs and KHIs [38–45]. ILs are salts that are generally composed of heterocyclic cations and inorganic anions [38]. ILs have been developed as both green chemicals and designer solvents since their structure and physical properties can be fine-tuned for diverse and specific applications [10,46,47]. ILs are water-soluble, non-volatile, and less toxic than methanol, and the thermodynamic inhibition performance of some ILs is comparable to that of methanol [19,20]. Therefore, ILs are good substitutes to methanol as THIs.

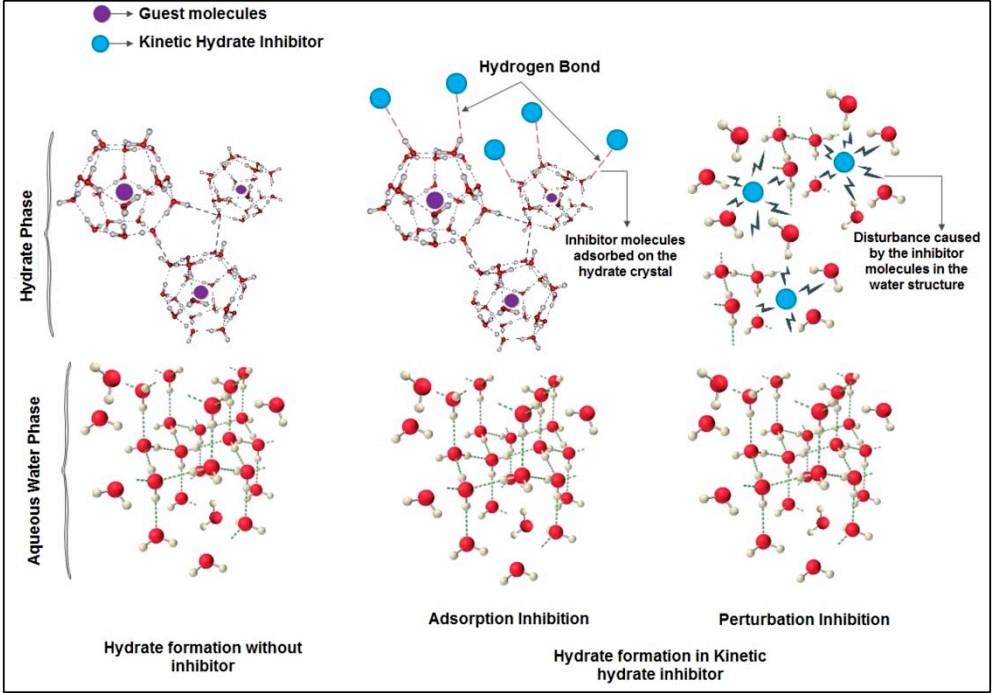

**Figure 2.** Schematic diagram of kinetic hydrate inhibition by adsorption and perturbation [36].

### 3.2. Quaternary Ammonium Salts

Quaternary ammonium salts (quats) (QAS) are one of the common economical IL compounds applied in industry [48]. Among their advantageous characteristics include having surface-active properties, bioactive, and possess anti-microbial activity [49,50]. Over the last decade, researchers have been investigating on this class of IL and found their improved chemical and thermal stability compared to imidazolium and pyridinium based ILs, their solvating properties and unique miscibility which can be utilized in specific applications [45,51–54]. A typical QAS consists of a positively-charged nitrogen atom attached to four carbon atoms [55]. Figure 3 shows a typical structure of QAS, tetramethyl ammonium chloride.

**Figure 3.** Molecular Structure of tetramethyl ammonium chloride.

Table 1 shows a summary of several studies conducted on QAS for gas hydrate inhibition and a brief findings on their inhibition performance. In evaluating the performance of QAS for gas hydrate inhibition, several methods has been applied, such as the T-cycle method and using highpressure micro differential scanning calorimetry equipment. The Isochoric constant cooling method or T-cycle method is one of the most used method, carried out by [37,56–62] with different type of gases to study on the kinetic inhibition of the selected compounds. In this method, 100 mL of liquid sample is poured into the cell which is then vacuumed thoroughly to remove excess air that may still exist in the cell [20,63]. The initial gas temperature is set to a temperature above the hydrate equilibrium temperature to avoid gas hydrate formation during pressurization. Then, the cell was pressurized to the specified experimental pressure. The reactor was kept for stabilization until the pressure reaches at its equilibrium with the experimental condition. The pressure and temperature measurements start getting logged when the temperature begins to decrease by rapid cooling method to facilitate the hydrate formation as it quickly rises to the desired experimental temperatures [20]. The pressure and temperature profiles are recorded for every 10 s through data acquisition software.

**Table 1.** Summary of studies conducted on ammonium based ionic liquids for gas hydrate inhibition.

| Chemicals | Chemical Formula | Conc. | Operating Condition | Method & Type of Gas | Application as | Main Findings | Year/Ref. |
|---|---|---|---|---|---|---|---|
| Tetraethylammonium iodide (TEAI) | $C_8H_{20}IN$ | 1 wt% 5 wt% 10 wt% | 274–284.6 K 3.45–8.3 MPa | T-cycle method $CH_4$ | THI | The suppression temperature of TMAB, TEAB, and TEAI at 10 wt% is 1.34 K, 1.07 K, and 0.82 K, respectively. TMAB performed better than TEAB and TEAI, individually and in combination with MEG. | 2020 [58] |
| Tetramethylammonium bromide (TMAB) | $C_4H_{12}BrN$ | | | | | | |
| Tetraethylammonium bromide (TEAB) | $C_8H_{20}NBr$ | | | | | | |
| Tetramethyl ammonium chloride (TMACl) | $C_4H_{12}NCl$ | 1 wt% 5 wt% 10 wt% | 285.0 K and 8.00 MPa for $CH_4$ 283.0 K and 3.50 MPa for $CO_2$ | T-cycle method $CO_2$ & $CH_4$ | THI & KHI | TMACl performed efficiently as a potential dual functional hydrate inhibitor for both $CO_2$ and $CH_4$ gases. The average suppression temperatures for TMACl at 1, 5, and 10 wt% are 0.70 K, 0.96 K, and 1.42 K, respectively. | 2019 [37] |
| Tetraethyl ammonium iodide (TEAI) | $C_8H_{20}IN$ | 5 wt% 10 wt% | 275.0–283.0 K 2.0–3.50 MPa | T-cycle method $CO_2$ | THI | The suppression temperature of TEAI, TEAB, and TMAB at 10 wt% is 1.17 K, 1.22 K, and 1.57 K, respectively. TMAB performed better than TEAB and TEAI individually and in mixture with MEG. | 2019 [59] |
| Tetraethyl ammonium bromide (TEAB) | $C_8H_{20}NBr$ | | | | | | |
| Tetramethyl ammonium bromide (TMAB) | $C_4H_{12}BrN$ | | | | | | |
| Tetraethyl ammonium chloride (TEACl) | $C_8H_{20}ClN$ | 10 wt% | 272.65–298.15 K 4.1–7.1 MPa | Isochoric pressure search $CH_4$ | THI & KHI | TEACl enhances methane hydrate storage capacity and reduce methane hydrate stability. | 2019 [60] |
| Tetraethylammonium chloride (TEACl) | $C_8H_{20}ClN$ | 4.77 wt% 9.15 wt% 11.82 wt% | 274.6–283.4 K 3.18–7.93 MPa | Isochoric pressure search $CH_4$ | THI | Addition of 11.82 wt% TEACl and 11.82 wt% of BMIM-$BF_4$ mixture results in more reduction in methane hydrate equilibrium temperature (average temperature depression of 2.7 K), compared to the other two studied mixtures. The inhibition effect is also enhanced when the system pressure is increased. | 2019 [57] |
| Tetramethyl ammonium bromide (TMAB) | $C_4H_{12}BrN$ | 0.05 and 0.1 mass fraction | 282.4–276.8 K 4.2–7.6 MPa | Isochoric pressure search $CH_4$ | THI | TMAB and TEAB show hydrate formation inhibition effects thermodynamically. However, TBAB has shown methane hydrate promotion effect. | 2018 [56] |
| Tetraethyl ammonium bromide (TEAB) | $C_8H_{20}NBr$ | | | | | | |
| Tetrabutyl ammonium bromide (TBAB) | $C_{16}H_{36}BrN$ | | | | | | |
| Tetraethylammonium iodide (TEAI) | $C_8H_{20}IN$ | 0.1 mass fraction | 5.1–11.1 MPa | HighPressure Micro DSC $CH_4$ | THI | The presence of TEAI alters the HLVE boundary to a higher pressure and lower temperature. | 2018 [64] |
| Tetra-n-butylammonium bromide (TBAB) | $C_{16}H_{36}BrN$ | 15,000 ppm | −0.5 °C | Isothermal test and maximum subcooling test THF | THI & KHI | Crystal growth inhibition is the dominant inhibition mechanism in the gas hydrate system operating in these mixtures. Although THAB demonstrated poor inhibition effect with PVCap in the THF hydrate tests, it shows synergy with the gas hydrate system. | 2017 [65] |
| Tetra-n-butylphosphonium bromide (TBPB) | $C_{16}H_{36}P·Br$ | 4500 ppm | | | | | |

**Table 1.** *Cont.*

| Chemicals | Chemical Formula | Conc. | Operating Condition | Method & Type of Gas | Application as | Main Findings | Year/Ref. |
|---|---|---|---|---|---|---|---|
| Tetramethyl ammonium bromide (TMAB) | $C_4H_{12}BrN$ | 10 wt% | | | | | |
| Tetraethyl ammonium bromide (TEAB) | $C_8H_{20}NBr$ | 10 wt% | 278.94–291.85 K 4.79–14.32 MPa | Isochoric pressure search $CH_4$ | THI | TMAB, TEAB, or TPrAB slightly alters the phase equilibrium conditions to a lower temperature and higher pressure region, which is comparable to NaCl. In contrast, the addition of TBAB and TPeAB promotes hydrate formation. | 2016 [61] |
| Tetrapropyl ammonium bromide (TPrAB) | $C_{12}H_{28}BrN$ | 10 wt% | | | | | |
| Tetrabutyl ammonium bromide (TBAB) | $C_{16}H_{36}BrN$ | 10 wt% 5 wt% | | | | | |
| Tetramethylammonium bromide (TMAB) | $C_4H_{12}BrN$ | 0.62 mol% | | | | | |
| Tetraethylammonium bromide (TEAB) | $C_8H_{20}NBr$ | 0.62 mol% | | | | | |
| Tetrapropylammonium bromide (TPrAB) | $C_{12}H_{28}BrN$ | 0.62 mol% | 279.41–291.85 K 4.79–14.32 MPa | Step-heating pressure search method $CH_4$ | THI & KHI | TBAB or TPeAB shows semiclathrate hydrate promotion effect. TMAB, TEAB or TPrAB shows slight inhibition effect. | 2016 [62] |
| Tetrabutylammonium bromide (TBAB) | $C_{16}H_{36}BrN$ | 0.62 mol% | | | | | |
| Tetrapentylammonium bromide (TPeAB) | $C_{20}H_{44}BrN$ | 0.62 mol% | | | | | |

In order to establish the effects of additional chemicals and salts on water activity, there are different types of thermodynamic models available, such as the Dickens and Quinby Hunt [58,66] which is useful in determining the hydrate liquid vapor equilibrium for $CH_4$ hydrates. This model has also been used by other researchers [67,68] due to its accuracy in predicting the hydrate phases stability boundaries [69] in the presence of IL solutions and amino acids. In evaluating the performance of a thermodynamic hydrate inhibitor, the average suppression temperature, $\Delta T$ is one of the key indicator [70]. The suppression temperature ($\Delta T$) is calculated by the following formula (1) [20].

$$T = \frac{\Delta \text{T}}{n} = \frac{\sum_{i=1}^{n}\left(T_{0,\,pi} - T_{1,\,pi}\right)}{n} \tag{1}$$

where $T_{0,\,pi}$ is the temperature at equilibrium conditions for gas in a blank sample which does not contain any compound. The equilibrium temperature of gas containing inhibitor is represented by $T_{1,\,pi}$. In both cases, the dissociation temperatures must be determined at the same pressure values. Symbol 'n' denotes the number of points for pressure considered in the experimentation. [58]. In researches conducted by [58,59] using different types of gases, they demonstrate that as the molecular weight of the IL reduces, the average suppression temperature, $\Delta T$ increases. The average suppression temperature when inhibitor is tested with $CO_2$ gas is better compared to $CH_4$ gas since it gives a slightly higher value, as shown in Figure 4.

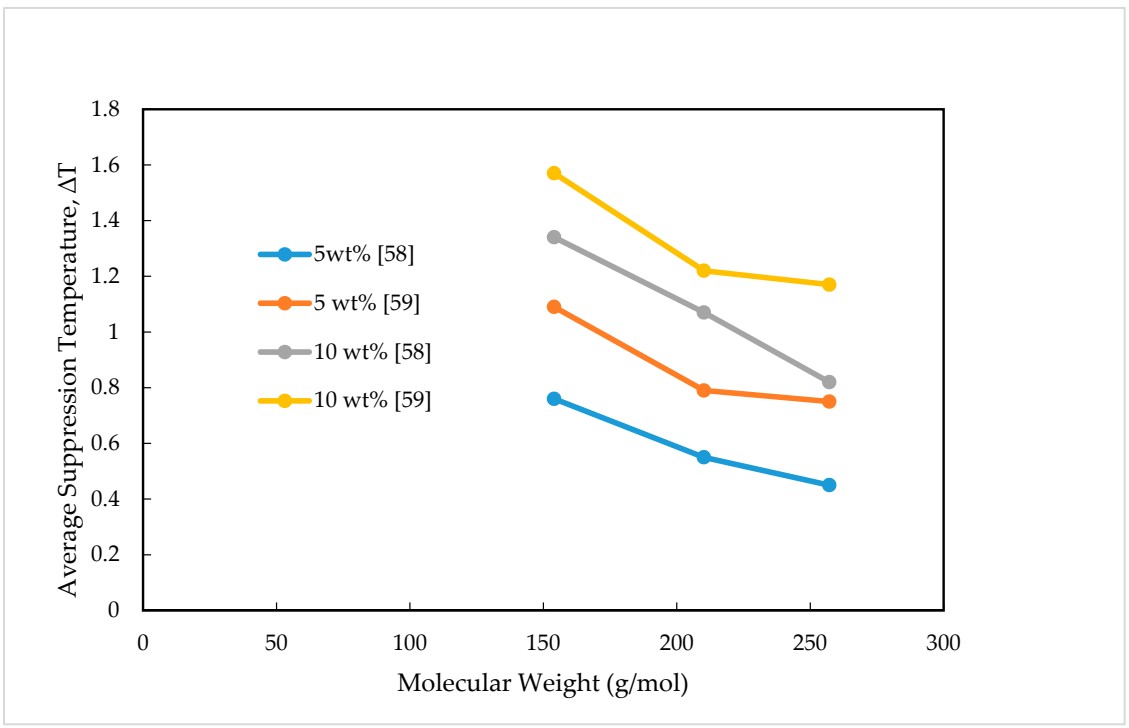

**Figure 4.** Average Suppression Temperature at 5 wt% and 10 wt% for Different Molecular Weight of AmmoniumBased ILs.

It has been suggested that anions of ILs play a leading role in inhibiting gas hydrates thermodynamically, while cations can also contribute to gas hydrate inhibition when specific functional groups, like a hydroxyl group (-OH), are added to them, or their chain length is modified, thus making them able to interact with water molecules [29,31,38,71]. Furthermore, the thermodynamic gas hydrate inhibition efficiency for ILs with the same anion attached to shorter alkyl chain ($C_2$) substituent is better than that of ILs with longer alkyl chain ($C_4$) substituent [10,72]. This indicates that the length of alkyl chain [73] is one of the factors that contributes to the performance of hydrate inhibitor. Larger alkyl chain carries hydrophobicity and prohibits cations in ILs to have a bond with the hydroxyl

ion [58]. In studying the effect of IL cations, the efficiency of ILs as THIs reduced with increasing chain length of the cation [43,74]. This is further evidenced by a study conducted by [20] using several ammonium based ILs at 10 wt% by maintaining the same anion, where it shows a similar trend of increasing average suppression temperature as the molecular weight decreases. It is also observed that the concentration of ILs is directly proportional to the average suppression temperature, as shown in research done by [37]. Inhibition performance is also enhanced when higher concentrations of ILs are used.

$\Delta H_{diss}$ is referred to the gas hydrate dissociation enthalpy. Dissociation enthalpies is calculated by employing Clausius-Clapeyron equation. For this purpose, derivative of the experimental HLVE data is obtained. Dissociation enthalpy is calculated by finding slope of hydrate liquid vapor equilibrium data using the Clausius-Clapeyron equation as the following formula (2) [58].

$$\frac{\mathrm{d}\ln P}{d\frac{1}{T}} = \frac{\Delta H_{diss}}{zR} \tag{2}$$

where in $T$ is the equilibrium temperature and $P$ is the pressure at which equilibrium temperature is inspected. Universal gas constant is presented as $R$ and $z$ is the compressibility factor of the gas used [58]. Hydrate dissociation enthalpy, $\Delta H_{diss}$ is dependent upon two main factors which are, the ability of clathrate structure to form hydrogen bond and cage occupancy of gas molecules [75]. Since the dissociation enthalpy values do not significantly change, it can be concluded that the chemical compounds used as inhibitor did not contribute significantly in the hydrate crystallization phase [58].

Other than average suppression temperature and hydrate dissociation enthalpy, the performance of hydrate inhibitor is also evaluated on its electrical conductivity. Recently, Ref. [56] have studied on the effects of tetrabutyl ammonium bromide (TBAB), tetraethyl ammonium bromide (TEAB), and tetramethyl ammonium bromide (TMAB), [56] on inhibiting $CH_4$ gas hydrate formation by isochoric pressure search method. The experiment was conducted at temperature range of 282.4–276.8 K and pressure range of 4.2–7.6 MPa by varying concentrations, i.e., 0.05 and 0.10 mass fraction. It is observed that the addition of TMAB and TEAB shifts the phase equilibrium curve of $CH_4$ gas hydrate to lower temperature and higher pressure regions, whereas the addition of TBAB demonstrates characteristic of hydrate formation effect. Electrical conductivity experiments results indicate that the shorter alkyl chain length of the ammonium ILs yield to higher electrical conductivity, and the sequence of electrical conductivity is TMAB > TEAB > TBAB [56]. The electrical conductivity is also found to be directly proportional to the concentration. In general, ILs with higher electrical conductivity show higher thermodynamic hydrate inhibition effects [43,56].

Recently, Fatemeh et al. [57] have studied on the effects of QAS and IL, BMIM-BF$_4$ on the thermodynamic stability via experiment (isochoric pressure-search method) and modeling. The van der Waals-Platteeuw (vdWP) theory is the main reference in calculating the chemical potential of water in hydrate phase [76]. On the other hand, the Peng-Robinson (PR) equation of state [77,78] is referred to calculate the fugacity in the gas phase. The NRTL activity coefficient model [79,80] is also used to investigate the water activity in the aqueous phase. It was found that the inhibition performance of methane hydrate is enhanced when both QAS and IL is combined, in which, it shows average temperature depression value of 2.7 K, when compared to its individual performances, with 0.8 K and 1.0 K for 25 wt% of TEACl and 25 wt% of BMIM-BF$_4$, respectively.

## 4. Conclusions

This paper covers on different types of QAS for gas hydrate inhibition for different type of gases at variety of operating conditions, concentrations, and by using different methods. It also gives a correlation of length of alkyl chain, average suppression temperatures, hydrate dissociation enthalpies, and electrical conductivity [81] to the effectiveness of the QAS as gas hydrate inhibitors. The QAS exhibited hydrate inhibition characteristics that are similar to the traditional thermodynamic hydrate inhibitors such as methanol and glycol [9], in which they shift the hydrate liquid vapor equilibrium

curve to a lower temperature and higher pressure. It is recommended that future research works to be conducted in investigating the synergy effects of QAS with other compounds, via chemical bonding interactions, such as amino acids, fatty acids and other types of polymers.

**Author Contributions:** Conceptualization, H.H.H. and H.H.; Resources, H.H.H. and H.H.; Writing—Original Draft Preparation, H.H.H.; Writing—Review and Editing, H.H.H. and H.H. All authors have read and agreed to the published version of the manuscript.

**Funding:** This research work conducted by the first author was funded by YUTP-FRG grant (015LC0-064).

**Acknowledgments:** The authors would like to acknowledge and thank the Department of Petroleum Engineering UTP and YUTP-FRG grant (015LC0-064) in supporting this research.

**Conflicts of Interest:** The authors declare no conflict of interest.

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
