# Peer review of "Review on Application of Quaternary Ammonium Salts for Gas Hydrate Inhibition"

_applsci, doi:10.3390/app10031011_

Round 1

Reviewer 1 Report

This paper covers the latest types of ionic liquid hydrate inhibitor (2017 – 2020) and a summary of findings which are essential for future studies. This paper also covers different types of ammonium-based ionic liquids for gas hydrate inhibition for a different type of gases at a variety of operating conditions, concentrations, and by using different methods. Although this categorization can potentially provide interesting results around this subject area, it has not been sufficiently addressed. From a review paper, it is expected to justify the literature gap and present some new perspective or categorization to its audience. Referring to these concerns and in accordance with the lack of novelty, I cannot recommend this paper for being published in this journal. The authors can highlight their points after a major revision. 

Author Response

Dear Reviewer 1:

Thank you.

Best Regards,

H.H. Hussain

Reviewer 2 Report

This manuscript discusses the use of alkylammonium salts as potential inhibitors for gas hydrate formation.  While the science of gas hydrate inhibition is of interest both fundamentally and for applications, this manuscript provides no new contribution to this science.

The authors refer to the family of alkylammonium salts as ionic liquids, even referring to tetramethyl ammonium as a typical ammonium based IL.  TMAC exhibits a melting point of 425 °C, and by no definition that I am aware of could be classified as an ionic liquid.  The authors are at best evaluating various families of alkyl ammonium salts, not ionic liquids.

The ionic liquid nomenclature problem is exaggerated throughout the claims in section 3.2 where the authors recite common overgeneralizations of the wonders of ionic liquids.  Where potentially exhibiting green applications, IL’s value is generally a result of their low volatility, which would not be a significant factor in such gas-hydrate inhibition applications.  Furthermore, these materials only exhibit properties as ionic liquids when they are the primary species, i.e. the solvent.  For gas hydrate inhibition, these materials are dissolved and at low concentrations, thus no-longer exhibit their ionic liquid properties.  They are simply dissolved ions.  That said, there may be some interesting correlations between ions that form ionic liquids and those same ions ability to inhibit gas hydrate formation.  But this is not discussed.

Table 1 is the major data table of this manuscript.  But notably the column of “main findings” presents at best qualitative descriptions.  What is the data used, or reference frame to determine increased or decreased abilities to form hydrates.  At a minimum it would seem they should present data such as for equations 1 and 2.

The authors note an apparent molecular weight effect on the gas-hydrate suppression temperature.  This of course is no surprise.  They report only wt. % for the use of the inhibitor.  For a given wt. %, a material with a smaller MW will have more moles of solute dissolved, thus, exhibiting greater colligative effects.  Without comparison of mole fractions no useful scientific conclusions can be made with regard to MW, chain length, or functional group dependencies.

Overall, this manuscript provides no new information to the field, and the information provided is not sufficient to make any scientific determinations.  Thus I cannot recommend publication.

Author Response

Dear Reviewer 2:

Thank you.

Best Regards,

H.H. Hussain

Reviewer 3 Report

This is a useful addition on the topic of Gas Hydrate Inhibition based ionic liquids and in particular the properties of quaternary ammonium salts and deserves publication.

Here are some minor revisions below:

Page 2: Figure 1, (the resolution is not good and it should be improved) 

Page 4,

Table 1, (you have to mention the type of ILs application as KHIs and/ or THIs, so I recommend to add a column for application types). Table 1, please keep the consistency of writing the abbreviation of ILs, e.g. C8H20IN should be C8H20NI. Table 1, Keep a space or use a dashed line between each ILs and references. Table 1, You have to decrease the font size of Table 1 in order to get the complete expression. Page 4, Section 3.2. Ammonium Based Ionic Liquids,

it will be worthy of this review if you can add a figure shows how QAs-ILs inhibit the gas hydrate formation.

Author Response

Dear Reviewer 3:

Thank you.

Best Regards,

H.H. Hussain

Round 2

Reviewer 1 Report

The authors have reasonably addressed my comments. I am happy for this paper to be published as it is in present form.